

# Knowledge and attitude of dental school faculties towards stem cell therapies and their applications

Sumaiah Ajlan and Nahid Ashri

Department of Periodontics and Community Dentistry, King Saud University, Riyadh, Saudi Arabia

## ABSTRACT

**Background**. Stem cells are characterized by their ability to self-renew and differentiate across multiple lineages. Knowledge about these cells and their potential is an important factor driving people to support their use. Therefore, we aimed to evaluate the knowledge and attitudes regarding stem cell therapies, education, and donation among faculty members of the College of Dentistry at King Saud University, Saudi Arabia.

**Methods**. A self-completed questionnaire was distributed among the faculty members, in which questions about demographic data, basic knowledge of stem cells as well as attitudes towards education and participation in stem cell-related research, and tissue donation were raised.

**Results**. A total of 102 participants responded to the questionnaire. Most participants (62.7%) ranked their stem cell knowledge as basic, despite correctly answering most of the knowledge questions. More than half of faculty mentioned a lack of stem cell-related activities in their courses (59.8%), and most showed strong support for including related education in postgraduate programs (71.6%). Noteworthy, 64% of participants reported interest in stem cell research and 74.5% were willing to donate their tissue for research purposes. The mean ranks of knowledge scores were significantly higher in subjects who have related lectures in undergraduate courses ($p < 0.05$), and those with high interest in research participation and tissue donation ($p < 0.0001$), and ($p < 0.05$) respectively. Most participants ($>60\%$) were not aware about the current applications of stem cell research and therapy in Saudi Arabia.

**Conclusion**. Most participants had an acceptable degree of knowledge about stem cells and showed a positive attitude to their education and participation in research. Additionally, knowledge scores have significantly and positively influenced those attitudes. More effort is required to promote the presence of local facilities for researchers to participate in stem cell-related research. The results of this study can be used as a foundation for evidence-informed policymaking within dental schools and research institutions.

# INTRODUCTION

Stem cells are a type of progenitor cell characterized by self-renewal and multilineage differentiation capabilities (*Zakrzewski et al., 2019*). Different types of stem cells exist and

Corresponding author
Sumaiah Ajlan, Sajlan@ksu.edu.sa

can be categorized according to their origin, for example embryonic stem cells, that are derived from growing blastocysts, or adult stem cells obtained from other body parts. Adult stem cells can be further categorized into hematopoietic cells, which are derived from bone marrow, or mesenchymal cells derived from multiple tissues (*Zakrzewski et al., 2019*).

Because of their characteristics, stem cells have been considered as a reservoir of reparative cells that would be useful for various medical applications, including enhancement of tissue healing (*Imam & Amer, 2023*; *Wu et al., 2022*), improvement of regeneration (*Qasim, Chae & Lee, 2020*; *Wrzyszcz-Kowalczyk et al., 2021*), and facilitation of tissue and organ bioengineering (*Basler et al., 2020*; *Qasim, Chae & Lee, 2020*; *Shahraki et al., 2022*). As such, stem cell research is among the most promising fields in medicine.

Recently, researchers have successfully used stem cell technologies for the management of diseases with previously poor prognoses, including the use of stem cells for certain life-threatening hematological and bone marrow malignancies (*Goel et al., 2021*), and blood disorders (*Lattanzi et al., 2021*; *Tayebi et al., 2017*). Other examined applications include the management of neurological disorders (*Zhang et al., 2020*), cardiac diseases (*Chen et al., 2022*; *Liu et al., 2022*), immunological disorders (*Cao et al., 2017*; *Puyade et al., 2022*), metabolic disorders (*Huang, Huang & Liu, 2021*) and dental regeneration (*Hu, Liu & Wang, 2018*; *Nazhvani et al., 2021*) among others.

Stem cells can often be isolated from tissues that are normally discarded, such as the umbilical cord, fatty tissue, and exfoliated/extracted teeth. Additionally, using cryopreservation, these cells can be preserved for future use with no significant changes in their characteristics if certain protocols are followed (*Alsulaimani et al., 2016*; *Cottle et al., 2022*; *Xie et al., 2022*; *Zhang et al., 2022*) . As a result, it is important to promote the donation of these tissues so that they can be used by others in need and preserved for potential future use by the donor. Knowledge of these cells and their enormous potential applications is one of the most critical factors in encouraging people to accept, participate in, and support their use. One of the primary barriers to individual participation in such research and tissue donation, according to a study conducted in Italy by *Conte et al. (2024)*, is a lack of appropriate knowledge. Interestingly, a lecture on hematopoietic stem cell transplant has enhanced the knowledge and increased the willingness for their donation (*Kaya et al., 2015*). Similar findings were also reported from local studies by other investigators (*Al-Shammary & Hassan, 2023a*; *Alsalamah et al., 2023*; *AlSubaie et al., 2023*; *Basudan et al., 2023*).

As such, many researchers have attempted to evaluate the knowledge and attitudes toward stem cell therapy and its application, among different population categories including university students and graduates (*Alhadlaq et al., 2019*; *Hemmeda et al., 2023*; *Hrenczuk, Gruszkiewicz & Malkowski, 2021*; *Luo et al., 2021*; *Narayanan et al., 2016*), healthcare providers (*Goswami, Kumar & Sharma, 2020*; *Katge et al., 2017*; *Kheirallah et al., 2021*), parents (*Peberdy et al., 2018*; *Szubert et al., 2020*), and the general population (*Alsalamah et al., 2023*; *Hamed et al., 2022*; *Korean Movement Disorders Society Red Tulip Survey et al., 2014*), and presented various results.

Health science students are the future physicians and medical staff, and are expected to further train the community and to spread knowledge about stem cell therapy and

their applications by providing the correct information and guiding their patients for the reliable use of their latest innovations (*Abdulrazeq et al., 2022*). Accordingly, several authors tried to evaluate their level of knowledge about this technology (*Abdulrazeq et al., 2022*; *Hemmeda et al., 2023*) and promoted the inclusion of related activities in their educational courses (*Luo et al., 2021*; *Phelan & Szabo, 2019*; *Xu et al., 2024*). For example, research on Sudanese students (*Hemmeda et al., 2023*) reported a good level of knowledge regarding stem cell-basics including their division and self-renewal capabilities but they lacked essential expertise in some other aspects. This finding calls for more effort to be put into incorporating related educational activities into their curriculum.

In Saudi Arabia, several studies have evaluated health care students' knowledge and attitudes toward stem cell use. Most authors observed an adequate knowledge, with highly positive attitudes toward cell donation and research participation often indicating a gap in the making the most of this high level of interest (*Al-Shammary & Hassan, 2023a*; *Alhadlaq et al., 2019*; *Almaeen, Wani & Thirunavukkarasu, 2021*; *Hazzazi et al., 2019*). For instance, *Woodman et al. (2023)* found in a multicenter study that the knowledge of the stem cells among the students was fragmented, and stressed the importance of inclusion of this type of education in dental curriculum.

University plays a vital role in providing students with primary education and the knowledge that will apply throughout their professional lives. The American Dental Education Association (ADEA) has actively promoted curriculum reform and innovation, based on educational research, with the creation of the Commission on Change and Innovation in Dental Education (CCI) (*Pyle et al., 2006*). ADEA has renewed this commitment with the CCI 2.0 (*Feldman & Valachovic, 2017*).

Faculty members are the corner stone of the educational process and often serve as mentors for student research projects. Therefore, an evaluation of their own knowledge and attitudes was deemed necessary before planning such reforms. The stem cell-based research is associated with a high degree of controversy partly due to the perceived complexity of the subject and also due to the quick progress in this area. However, the sigma of stem cell research often originates from a lack of knowledge. Nevertheless, there is a deficiency in research results that deal with the experiences of dental educators in teaching stem cell-related subjects. The proper evaluation of faculties' knowledge, viewpoints, interests, and perceived difficulties can all be used to help identify ways to enhance dental education in this field (*Hoskin et al., 2019*). Additionally, faculty members can play an important role in leading research in these areas and mentoring student projects that encourages their future commitment to this kind of practice. Therefore, the identification of their perceived barriers is important for planning ways to manage them (*Chambers & Brady, 2021*).

Accordingly, this survey aimed to evaluate the knowledge of stem cell therapies and attitudes towards stem cell education, research participation, and tissue donation among faculty members of the College of Dentistry, King Saud University, Riyadh, Saudi Arabia. In this research, we hypothesize that faculty members have good knowledge of those therapies, and positive attitudes toward their education, study participation and tissue donation.

## MATERIALS AND METHODS

### Study design and setting

This short-term cross-sectional study was conducted to evaluate the knowledge and attitudes of faculty members of the College of Dentistry at King Saud University in Riyadh, Saudi Arabia to stem cell-related research.

### Sample size estimation

The sample size was calculated using the Cochrane's formula of sample size estimation ($n = z^2(pq)/L^2$), where $n$ is the minimum required sample size, $z$ is desired confidence interval (95%), $p$ is the proportion, $q = (1-p)$ and $L$ is the desired level of precision.

   We assume 75% of dental professionals were aware about stem cells, with a precision of $\pm 8\%$ for a 95% confidence interval, the required sample size was 113 subjects. As the data collection was through on-line, a total of 230 dental professionals were approached to participate in the study

### Ethical considerations

The study protocol was approved by the Institutional Review Board of King Saud University Medical City (IRB no. E-22-6999), achieved on (July 2022). A copy of the approval is attached in the Supplementary file S1. This study was conducted following the 1975 Helsinki declaration, as revised in 2008 and following amendments or comparable ethical standards. A written, signed, informed consent was attached to the questionnaire. The consent briefly explained the study objectives, design and the expected duration (Supplementary files S2, S3). Participants invited to fill up the form voluntarily and did not receive any compensation or rewards for participating in the study. The participant identity was not revealed and data management confidentiality was guaranteed.

### Data collection instruments

A cross-sectional study using a self-administered questionnaire was conducted among faculty members at the College of Dentistry, King Saud University. The evaluation took place around the end of academic year 2022–2023. The questionnaire was designed and distributed in the English language as it is the official teaching language of the dental school. Initially, the questionnaire validity was evaluated by consulting expert supervisors from the university. This was followed by a pilot study of questionnaire distribution conducted among ten members to examine its reliability. An electronic questionnaire was then developed and distributed to approximately 230 faculty members *via* a university-based email address.

   Next, a paper-based version of the same questionnaire was prepared and distributed manually to colleagues who did not complete the electronic form and found it more convenient to complete the paper questionnaires.

   The questionnaire was divided into four different sections (Fig. 1). Initially, the survey evaluated basic demographic information and asked about the participants' age, gender, nationality, educational background, and academic standing.

   Thereafter, questions directed at the evaluation of several areas of stem cell-related knowledge were developed and divided into questions evaluating basic scientific knowledge
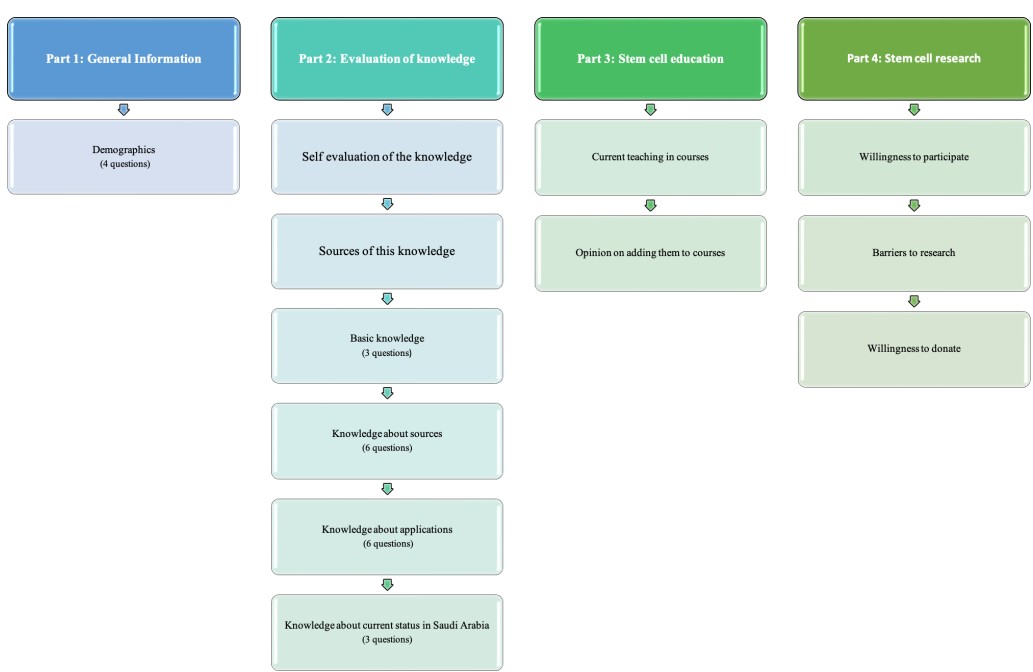

**Figure 1** Schematic representation of the questionnaire components.

about stem cells, including their definitions, fundamental standards, sources, and medical applications. Additionally, a question regarding the self-evaluation of overall knowledge was added. The primary sources of this knowledge were also evaluated. Questions evaluating the awareness of the state of stem cell therapy and research in Saudi Arabia were presented.

Subsequently, questions evaluating the current state of stem cell education at different student levels as well as the participants attitude towards the incorporation of stem cell-based education in student courses were included.

Thereafter, questions evaluating interest in participation in stem cell-based research and perceived barriers to its conduct were added. Finally, the questionnaire evaluated the participants' attitudes towards tissue donation for research purposes. A copy of the questionnaire is provided in the Supplementary file S4.

The percentage of participants with correct answers to the knowledge section was calculated and compared to the incorrect answers and neutral "not sure" answers. The overall personal scores were then correlated with the participants' demographic data, attitudes, and opinions. The internal consistency of the scale was assessed using Cronbach's alpha and its value was 0.926 (95% CI [0.903–0.945]). This is highly significant ($p < 0.0001$) and indicates that the items used in the questionnaire have good reliability (*Campbell, David & Stephen, 2021*).

## Statistical analysis

Data were analyzed using IBM SPSS Statistics software (version 26.0) for Windows (IBM Corp., Armonk, N.Y., USA). Descriptive statistics (median, interquartile range, frequencies,

and percentages) were used to describe skewed quantitative and categorical variables, as the knowledge scores were not following a normal distribution where Kolmogorov–Smirnov and Shapiro–Wilk tests $p$-values (0.049 and 0.005) indicates that the knowledge scores were deviating from normality. Hence the non-parametric statistical tests (Mann–Whitney U-test and Kruskal–Wallis test) were used to compare the mean ranks of knowledge scores of stem cell therapy and their application items in relation to the sociodemographic and professional characteristics of the study subjects. Statistical significance was set at $p < 0.05$ (*Campbell, David & Stephen, 2021*).

## RESULTS

### Response rate
Among the total number of submitted forms (230), a total of 102 of completed forms were received, yielding a response rate of 44.35%.

### Demographic characteristics and educational background
Most respondents were from Saudi Arabia (85.3%), with females representing slightly over 66% of the sample group (Table 1). Responses received were distributed among different age groups, with slightly higher response rates among participants >40 years old (36.3%) and 31–35 years old (35.3%). Approximately 28% of the respondents were from the prosthodontic department (Table 1).

### Assessment of stem cell-related knowledge
#### Basic knowledge
Most faculty ranked their stem cell knowledge as only basic (62.7%) (Fig. 2). However, their answers varied considerably, when asked specific questions regarding certain areas of knowledge (Table S1).

Regarding basic stem cell definitions, most faculty members answered two of the three questions correctly, with the percentage of correct answers being 80.4% and 63.7%, respectively. In contrast, the remaining question was answered "not sure" by many participants (42.2%).

With regards to the sources of stem cells, most participants were aware of cell isolation from the pulp of primary and permanent teeth, dental follicles, and periodontal ligament cells. This was indicated by the correct answers to related questions from the majority of participants, where the percentage of correct answers ranged from 52 to 57.8%. With the remaining two questions, most participants answered "not sure".

Regarding stem cell applications, three questions were correctly answered by most faculty members (50%, 55.9%, 52%), while the remaining questions were answered as "not sure". For all questions, the wrong answer was only chosen by the minority of the participants.

There were statistically significant differences with respect to age group, gender, and specialty when comparing the mean ranks of knowledge scores of stem cell therapies and their application items with the sociodemographic characteristics of study subjects. The mean knowledge score ranks were significantly higher in the age groups 24–30 and 31–35

**Table 1** Distribution of socio-demographic characteristics of study subjects ($n = 102$) and the comparison of mean ranks of knowledge score items in relation to those characteristics.

| Characteristics | No. (%) | Knowledge score | | p-value |
|---|---|---|---|---|
| | | Median (IQR) | Mean ranks | |
| Age groups (in years) | | | | |
|    24–30 | 13(12.7) | 28(8) | **61.88** | 0.002* |
|    31–35 | 36(35.3) | 29(8) | **62.82** | |
|    36–40 | 16(15.7) | 24.50(7) | 48.34 | |
|    >40 | 37(36.3) | 22.50(8) | 38.20 | |
| Gender ($n = 81$) | | | | |
|    Male | 27(33.3) | 29(10) | **49.81** | 0.017* |
|    Female | 54(66.7) | 25.50(7) | 36.59 | |
| Nationality | | | | |
|    Saudi | 87(85.3) | 26(9) | 52.18 | 0.573 |
|    Non-Saudi | 15(14.7) | 27(8) | 47.53 | |
| Specialty | | | | |
|    Maxillofacial surgery | 4(3.9) | 29(7) | **62.00** | |
|    Oral Diagnosis | 4(3.9) | 29.50(6) | **67.75** | 0.004* |
|    Endodontics & Restorative dentistry | 24(23.5) | 29(7) | **69.33** | |
|    Periodontics & Community dentistry | 22(21.6) | 27(7) | 54.18 | |
|    Prosthodontics | 28(27.5) | 25(6) | 37.29 | |
|    Pedodontics & Orthodontics | 8(7.8) | 23(9) | 40.63 | |
|    Others | 12(11.8) | 25(12) | 42.42 | |

**Notes.**
*Statistically significant.
Mean ranks of knowledge scores showing statistical significance are presented in bold.

years compared to those of the other age groups ($p = 0.002$) (Table 1). Male participants had significantly higher knowledge scores than female participants ($p = 0.017$). Finally, knowledge scores were significantly higher in subjects from the specialties of Maxillofacial surgery, Oral Diagnosis, and Endodontics & Restorative dentistry than in subjects from other specialties ($p = 0.004$). In terms of the knowledge scores, no statistically significant differences were observed regarding the nationalities of the study participant (Table 1) or in relation to self-evaluation of the knowledge (Fig. 2).

### Knowledge about stem cell therapy in Saudi Arabia

Most participants (>60%) were not sure about the current applications of stem cell research and therapy in Saudi Arabia, while only 19.6% were certain about the availability of multiple stem cell research centers in Riyadh, and only 29.4% believed that research on embryonic stem cells is carried out locally. Additionally, only 29.4% of the respondents were certain that stem cell therapy was used for the management of leukemia in local hospitals (Table S1).

### Sources of stem cell knowledge

Regarding the sources of information on stem cells, more than half of the participants received the information from two or more sources among a total of nine source options

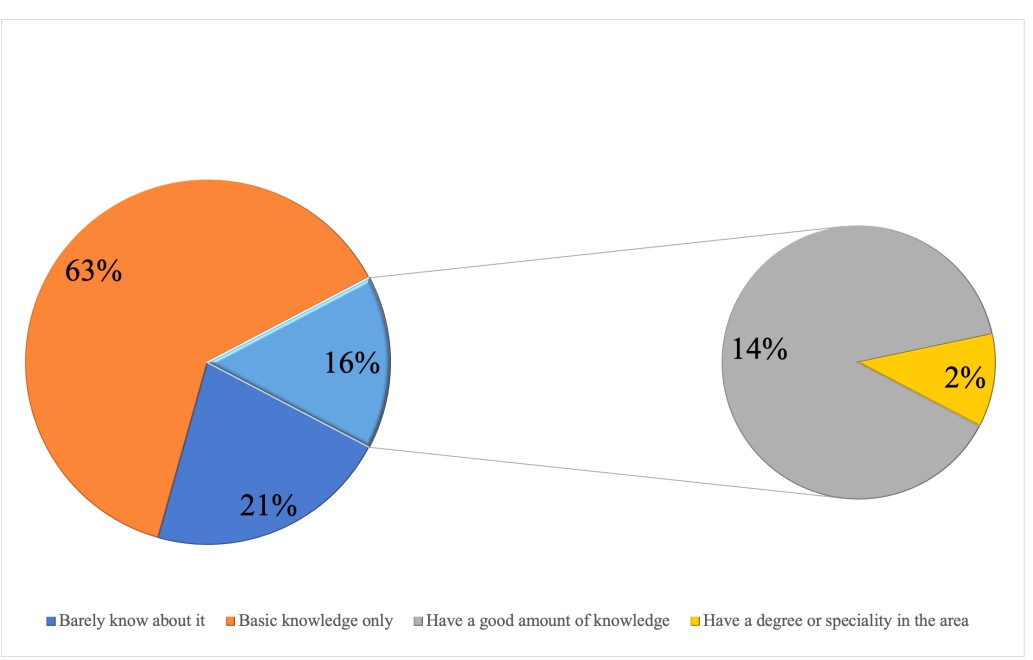

**Figure 2** Participants' self evaluation of stem cell related knowledge.

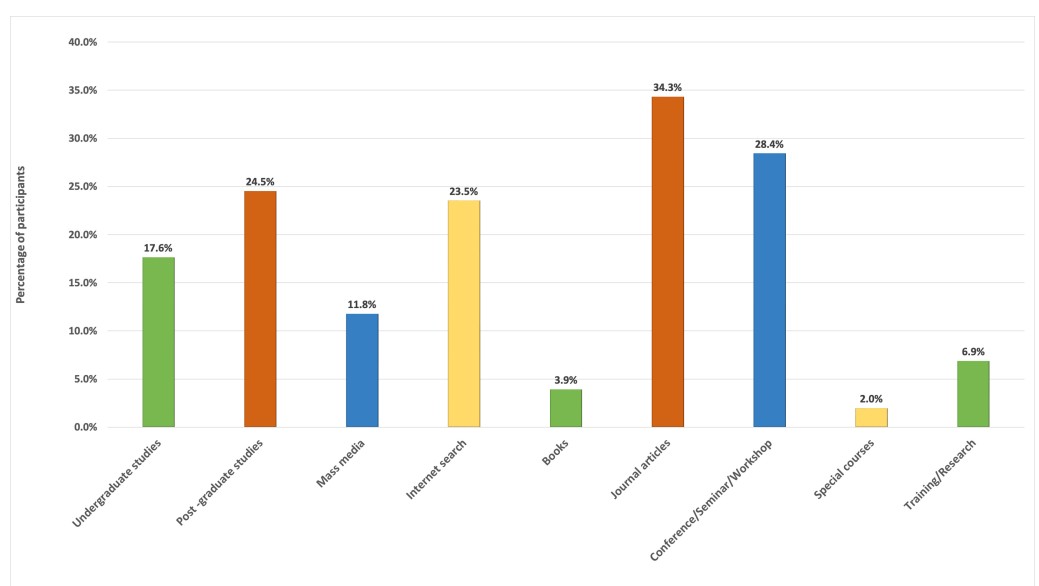

**Figure 3** Participants' responses regarding sources of stem cell related knowledge.

(undergraduate studies, post-graduate studies, mass media, internet, books, journal articles, conferences/seminars or workshops, special courses and training or research work). Further

**Table 2 Distribution of study subject's responses towards current status of stem cell education, interest in participation in stem cell research and in tissue donation with comparison of mean ranks of knowledge score items in relation to those responses.**

| Items for stem cell education, research participation, & tissue donation | No (%) | Knowledge score | | p-value |
|---|---|---|---|---|
| | | Median (IQR) | Mean ranks | |
| Any lectures contain stem cell information | | | | |
| Yes-Under graduate courses | 12(11.8) | 30(4) | **75.92** | 0.003* |
| Yes-Post graduate courses | 29(28.4) | 22(11) | 41.24 | |
| No | 61(59.8) | 26(6) | 51.57 | |
| Interested in participation in stem cell-based research | | | | |
| Yes | 65(63.7) | 28(6) | **60.62** | <0.0001* |
| No | 14(13.7) | 22.50(6) | 35.86 | |
| Not sure | 23(22.5) | 24(8) | 35.24 | |
| If you have removed/extracted any body tissue for medical reasons, would you agree to freely donate it for stem cell research? | | | | |
| Yes | 76(74.5) | 27(7) | **55.63** | 0.045* |
| No | 10(9.8) | 22.50(7) | 35.00 | |
| Not sure | 16(15.7) | 25.50(10) | 42.19 | |

Notes.
*Statistically significant.
Mean ranks of knowledge scores showing statistical significance are presented in bold.

analyses showed that, journal articles, conference seminars and workshops, post graduate studies and internet search were among the most frequently selected sources (Fig. 3).

## Stem cell education for students

The majority of faculty members expressed concern about the dearth of stem cell-related educational activities in their current courses when it came to incorporating stem cell therapy education for students, with only 41(40.2%) mentioning having related activities in their current courses, mostly postgraduate (Table 2). Mean ranks of knowledge scores of stem cell therapies and their application items were statistically significantly higher in subjects who included stem cell-related activities in their undergraduate courses compared with those who were only presenting related information during their postgraduate courses or those that were not providing related information at all, ($p = 0.003$).

When questioned about their opinion, most participants strongly agreed that stem cell education should be incorporated into postgraduate courses (71.6%). However, few participants (around 30%) strongly agreed that stem cell-based lectures and activities should be included in undergraduate courses or as extra-curricular activities (Fig. 4).

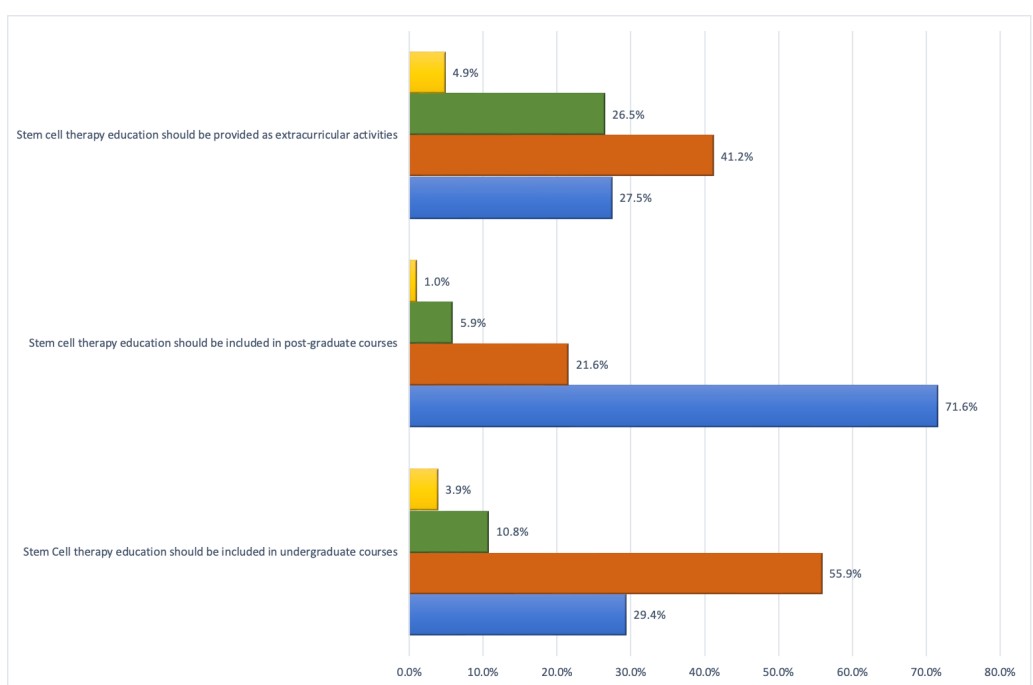

**Figure 4 Participants' opinions regarding regarding stem cell education.**

### Stem cell-related research and perceived barriers

About sixty-four percent of the participants responded positively to the question about interest in participating in stem cell-based research (Table 2). Subjects who reported being interested in research had significantly higher knowledge scores than those who were 'not interested' or 'not sure' ($p < 0.0001$) (Table 2).

For the possible barriers to perform related research, response proportions were used to rank the nine potential barriers to conducting stem cell-based research. The first barrier was ranked as ''lack of well-trained technicians'', the second as ''difficulties in accessibility to research center'', and the third as ''lack of proper facilities''. Six more barriers were then ranked (Table 3).

### Attitude toward tissue donation for research purposes

When asked about willingness to donate tissue samples for stem cell research, most participants were willing to freely donate their own tissue (74.5%), while only 15.7% were hesitant to donate (Table 2). The mean ranks of knowledge scores were significantly higher in subjects willing to donate tissue samples compared with those who were unwilling or unsure (Table 2).

## DISCUSSION

In this study, the response rate to the questionnaire was 44.35%. Earlier studies have shown that the response rates to questionnaires can be influenced by multiple factors, including
**Table 3  Distribution of ranking of responses towards the possible barriers for performing Stem cell-based research.**

| Barriers | Responses | | | | Ranking of barrier |
|---|---|---|---|---|---|
| | Strongly agree | Agree | Disagree | Strongly disagree | |
| Difficulties in accessibility to research center | 49(48.0) | 48(47.1) | 4(3.9) | 1(1.0) | 2 |
| Lack of proper facilities | 48(47.1) | 46(45.1) | 7(6.9) | 1(1.0) | 3 |
| Lack of well -trained technicians | 52(51.0) | 44(43.1) | 5(4.9) | 1(1.0) | 1 |
| Difficulty of obtaining materials | 39(38.2) | 50(49.0) | 12(11.8) | 1(1.0) | 5 |
| Time limitations | 26(25.5) | 36(35.3) | 37(36.3) | 3(3.9) | 7 |
| High Expenses | 35(34.3) | 47(46.1) | 19(18.6) | 1(1.0) | 6 |
| Ethical/ religious issues | 23(22.5) | 34(33.3) | 39(38.2) | 6(5.9) | 8 |
| The need for different type of training | 47(46.1) | 44(43.1) | 11(10.8) | – | 4 |
| Lack of interest | 14(13.7) | 37(36.3) | 40(39.2) | 11(10.8) | 9 |

study titles and length (*Lund & Gram, 1998*). Due to the nature of the topic, the title of the study may have given the impression of difficulty and affected participation.

The evaluation of the effect of demographic data at knowledge levels showed various results. For example, in the current study, male participants and those younger than 35 years old demonstrated better knowledge, which is in contrast to a previous study by Alzahrani that reported higher levels of knowledge and better attitudes toward therapy among female dentists, while age made no difference (*Alzahrani, 2019*). Similarly, *AlSubaie et al. (2023)* showed that females, younger individuals (18–25 years), healthcare workers, and those with higher education had better knowledge.

The questionnaire indicated a relatively acceptable level of basic knowledge among faculty members. High levels of awareness among dentists have already been reported (*Goswami, Kumar & Sharma, 2020*). This in turn can be due to the fact that the questionnaire was mainly answered by subjects interested in the field. *Al-Shammary & Hassan (2023b)* rated factors affecting knowledge among medical specialist and found that those with work experience scored higher on measures of sensitivity, acceptance attitudes, and knowledge. Similar findings were also reported by *AlSubaie et al. (2023)*.

Most faculties rated their stem cell knowledge as basic knowledge (62.7%), although they were able to answer many questions correctly. A tendency towards a lack of self-confidence in relation to the topic can be due to a lack of training in the field. Studies evaluating the accuracy of self-assessment have yielded mixed results, with a tendency towards underestimation of knowledge often associated with higher achievers (*Abeyaratne, Nhu & Malone, 2022*; *Jones, Panda & Desbiens, 2008*).

In this study the main sources of stem cell-based knowledge were identified as journal articles and conferences/seminars. The findings of *Sede, Audu & Azodo (2013)* evaluating Nigerian dentists further supports those results. Similarly, *Frati et al. (2014)* reported that approximately half of the physicians who demonstrated good knowledge of the topic had attended conferences/seminars or participated in specific training courses. However,

sources of knowledge differed significantly among other target groups, for instance healthcare science students (*Abdulrazeq et al., 2022*).

When questioned about their opinions on stem cell-related education for students, most participants agreed that only postgraduate courses should include stem cell educational activities. This finding may explain the results of a previous research by *Alhadlaq et al. (2019)*, who reported a low level of knowledge among recent graduates. Even though stem cell research is promising, there are still some unresolved issues and problems that could make people even less inclined to teach it in undergraduate courses (*Tatullo, 2018*). Meanwhile, initiatives are being implemented in multiple educational institutions to encourage undergraduate participation in modern and advanced research (*Gunn et al., 2018*; *Tavakol et al., 2018*). Several schools now include stem cell education for undergraduate students and present encouraging results (*Abdulrazeq et al., 2022*; *Kaya et al., 2015*; *Xu et al., 2024*). Recent local studies have shown a high level of awareness and a positive attitude towards the topic (*Almaeen, Wani & Thirunavukkarasu, 2021*; *Woodman et al., 2023*), a situation that justifies considering its inclusion for different student categories.

Overall, this study shows a high level of interest in research and activities related to stem cells, which is consistent with the findings of several other studies (*Luo et al., 2021*; *Sede, Audu & Azodo, 2013*). However, obstacles as lack of trained technicians, specialized research centers, and suitable machines and materials were mentioned as possible barriers discouraging participation in those research. Several studies have evaluated factors affecting faculties research productivity (*Chambers & Brady, 2021*; *Haden et al., 2010*; *Pau et al., 2017*) and identified time constrains and lack of proper training as major obstacles (*Chambers & Brady, 2021*). *Haden et al. (2010)* showed that an appropriately assigned budget and protected time are important factors for the participation of teachers in research.

Stem cell therapy has been used in Saudi Arabia since the 1980s, and some centers have used hematopoietic stem cell transplants to treat hematological diseases (*Abumaree et al., 2014*; *Alzahrani, 2019*). Although interest in stem cell-related therapy and research was initiated in a limited, unofficial manner, it has increased dramatically in recent years, with several specialized research centers established at various Saudi universities and hospital centers. Additionally, several stem cell-based studies from Saudi Arabia have been registered with the National Institute of Health (NIH) in recent years (*Abumaree et al., 2014*; *Alzahrani, 2019*). However, the current study revealed a general lack of awareness regarding stem cell therapy and its applications in Saudi Arabia. Such lack of awareness may further discourage researchers from taking part in those therapies.

An important aspect in the advancement of stem cell related research is the participation in tissue donation. A study by *Basudan et al. (2023)* have indicated variable attitudes of the public towards tissue donation for research purposes, where donation reluctance was associated with several factors including lack of proper knowledge. In our study when asked about willingness to donate tissue, most participants were willing to donate samples for these studies, indicating a high level of awareness of the importance of the donation among the targeted study population. This was expected due to the high level of education in the

selected study group. Attitudes toward donating certain tissue types have also been reported to be high among different study populations (*Narayanan et al., 2016*; *Screnci et al., 2012*). Launched by the King Abdullah International Medical Research Center (KAIMRC) in 2011, the Saudi Stem Cell Donor Registry (SSCDR) is the first of its kind in all Arab countries, has reached over 79,000 potential donors (*Abumaree et al., 2014*; *SSCDR, 2011*), which is promising and indicates the potential willingness of individuals to donate tissue samples. However, lack of proper knowledge about local agencies and donation possibilities can prevent the participation of possible donors, a finding that was repeatedly reported in several local studies (*Al-Shammary & Hassan, 2023a*; *AlSubaie et al., 2023*; *Jawdat et al., 2018*).

Actually, more thorough research is required to evaluate the topic's specific aspects; for instance, A study by *AlSubaie et al. (2023)* found that the presence of a family member in need of a transplant have enhanced the donation intentions. In contrast, the kinds of tissue requested, the difficulty of tissue procurement and the possible long term side effects may negatively impact the decision (*Hazzazi et al., 2019*; *Pinto da Silva et al., 2022*). Noteworthy, a study in Italy revealed that despite the general agreement among physicians regarding tissue donation, a high preference for tissue cryopreservation for future personal use was reported (*Frati et al., 2014*). Therefore, the presence of other options for using these tissues and the potential cost of the procedure can affect the participants' decision.

We believe that the survey results offer valuable insights, especially from a region where this area of research is still evolving. The findings of this study, for instance, may have an impact on the dental education policy making since it is commonly known that evidence-based practice and policy leads to better identification of deficiencies and proper use of resources. These findings can be taken into account, for example, when planning programs for faculty development at dental school. Based on the study's results, institutions should develop targeted educational programs and workshops to train faculty to utilize and apply stem cell therapies. In the same regards, *Iacopino (2007)* pointed to the importance of the continuous and timely incorporation of what he called "new science" into dental education so that dentistry can maintain its position as an essential component of the overall health care profession. According to him, this should be implemented during curriculum reform projects. The findings of our study encourage the inclusion of stem cell therapies and research in students' courses, as the relatively high level of knowledge of participants, coupled with high interest suggests that faculty members are ready for this step.

Additionally, our results underscore the need to assess further the barriers that may hinder the broader participation of researchers in this field. In fact, it is important that the administration recognizes and further evaluates the perceived obstacles if such research is to be promoted. In this regards, *Bland et al. (2005)* discussed the importance of creating a positive research culture in dental schools in order to enhance research productivity. This positive culture can include: time, resources, supportive personnel along with clear goals and communication techniques in a positive group climate and appropriate leadership skills. Moreover, this high research interest should encourage national and international collaborative work to improve the field.

Finally, the relatively low awareness about local stem cell applications indicates that more focus is required on the topic with better advertising of these centers in order to attract interested participants. Our findings indicate the universities' readiness to adopt broad policies to raise public awareness of the issue through community-based supervised educational initiatives.

Participants are advised to utilize their high interest and start to engage in related research. Additionally, they are encouraged to attend training courses related to the field in order to gain practical experiences. Also, faculty members are recommended to plan initiatives incorporating stem cell related knowledge and research to student courses. Finally, university faculties need to engage in public activities directed at raising the awareness among the society.

## CONCLUSION

Most participants had an acceptable degree of knowledge regarding stem cell therapies and their applications, and showed positive attitude towards education and participation in research as well as tissue donation. It was noted that knowledge has significantly and positively influenced those attitudes. Overall, our results indicate that more work is necessary to raise awareness of stem cell research and applications in Saudi Arabia in order to spread knowledge about the potential of stem cells and to encourage voluntary researchers to participate in stem cell-related studies. The results of this study can be used as a foundation for evidence-informed policymaking within dental schools and research institutions, and directly impact faculty development, curriculum reform, research initiatives, and clinical practices.

## SIGNIFICANCE OF THE STUDY

This study holds significant importance in advancing dental education and research by assessing the knowledge and attitudes of dental faculty members toward stem cell therapies. Faculty members play a critical role influencing the next generation of dental professionals, and their understanding of stem cell applications directly impacts curriculum development, research initiatives, and clinical practices. In fact, this study is crucial in shaping the future of dental education and research by addressing knowledge gaps, guiding curriculum reforms, promoting faculty engagement in research, and fostering a more informed approach to stem cell applications in dentistry.

## STRENGTH OF THE STUDY

This research evaluated the level of knowledge and the attitudes of dental school faculty on the stem cell education and applications. This group is very important and scarily evaluated in this area of research. The internal consistency evaluated using Cronbach's alpha coefficient and showed a high and significant level of consistency and good reliability (*Campbell, David & Stephen, 2021*). Accordingly, these statistics can enhance evidence-based policymaking among dental schools.

## STUDY LIMITATIONS/FUTURE DIRECTIONS

This study was conducted on a specific population from single institution which may reflect the point of view of this specific population and reduce generalizability of the findings. Secondly, the limited sample size and the response rate of 44.35%, indicates missing the perspectives of the remaining faculty members which could differ from those who responded. Additionally, the study design being cross-sectional has several drawbacks, for example, beside its limited identification for only the association between variables and failure to present causation, it presents the opinion at single time point and prevents dynamic understanding of evolving knowledge. Similarly, the reliance on self-administered questionnaires introduces the possibility of response bias. Participants may have overestimated or underestimated their knowledge and attitudes. Collectively, while the study identifies general trends, it does not provide qualitative insights into the reasons behind faculty members' attitudes, challenges they face, or their motivations for participation in stem cell research and education. Finally, the study lacked detailed evaluation of the barriers faced and the participants' suggested improvement plans.

More detailed studies are needed to evaluate the awareness and opinions of different sectors of society. Additionally, adopting longitudinal study designs revealing the change in knowledge and interest over time would be interesting. Multicenter studies reporting and comparing the opinions of larger number of faculty members can further enrich the literature. Adding qualitative components, such as focus groups or in-depth interviews, could provide deeper insights into faculty perceptions, barriers to participation, and potential solutions to improve engagement. Future studies should refine survey questions to assess more nuanced aspects of stem cell knowledge and attitudes. Including open-ended questions could allow participants to share more detailed perspectives. Conducting a separate study focusing on institutional challenges, such as funding availability, research infrastructure, and administrative support, would help identify actionable steps to enhance faculty involvement in stem cell research. Conducting parallel studies on dental students and the general public could provide a more comprehensive understanding of how stem cell education and awareness be improved at different levels.

By addressing these limitations and implementing the suggested improvements, future research can provide a more comprehensive and impactful understanding of stem cell education, research participation, and faculty engagement in dentistry.

**Abbreviations**

| | |
|---|---|
| **AADE** | American Dental Education Association |
| **CCI** | Commission on change and innovation in dental education |
| **IRB** | institutional review board |
| **IQR** | interquartile range |
| **KAIMRC** | King Abdullah International Medical Research Center |
| **NIH** | National Institute of Health |
| **SPSS** | Statistical Package for Social Science |
| **SSCDR** | Saudi Stem Cell Donor Registry |

## ACKNOWLEDGEMENTS

The authors would like to thank Prince Naif Health Research Center, Investigator support Unit for the language editing service provided.

### Funding

The authors received no funding for this work.

### Competing Interests

The authors declare there are no competing interests.

### Author Contributions

- Sumaiah Ajlan conceived and designed the experiments, performed the experiments, analyzed the data, prepared figures and/or tables, authored or reviewed drafts of the article, performed pilot study/distributed the questionnaire, and approved the final draft.
- Nahid Ashri conceived and designed the experiments, authored or reviewed drafts of the article, communicate with respondents, and approved the final draft.

### Human Ethics

The following information was supplied relating to ethical approvals (i.e., approving body and any reference numbers):

The study protocol was approved by the Institutional Review Board of King Saud University Medical City.

### Ethics

The following information was supplied relating to ethical approvals (i.e., approving body and any reference numbers):

Institutional Review Board of King Saud University Medical City.

### Data Availability

The raw data are available in the Supplementary Files.

### Supplemental Information

Supplemental information for this article can be found online at http://dx.doi.org/10.7717/peerj.19127#supplemental-information.

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
