# Peer review of "Knowledge and attitude of dental school faculties towards stem cell therapies and their applications"

_PeerJ, doi:10.7717/peerj.19127_

## Round 0.1 · original submission · Major Revisions

Your manuscript has been reviewed and assessed by three reviewers, and there are still a few points that need to be addressed. The comments of the reviewers are included at the bottom of this letter. We would be glad to consider a substantial revision of your work, where the reviewer’s comments will be carefully addressed one by one. The manuscript is a valuable contribution to understanding the knowledge and attitudes of dental faculty toward stem cell therapies. It is written in clear, professional English and provides adequate background and context, emphasizing the importance of stem cell knowledge in dental education and research. However, several improvements are suggested to enhance the manuscript's clarity, methodological rigor, and presentation:

Language and Clarity: While the manuscript is generally well-written, some sentences in the "Results" section could be more concise. Additionally, the introduction should be elaborated to justify the study more comprehensively, supported by additional references and hypotheses.

Methodological Details: The time period of data collection should be added. The Cronbach’s Alpha value for the questionnaire should be reported to validate its reliability and internal consistency. The choice of non-parametric tests should be clarified by including normality assumption testing for added transparency. Details on sample size calculation, software used, and questionnaire calibration by experts should be provided. The inclusion of the questionnaire as an additional file is recommended.

Data Presentation: The tables are well-labeled and effectively summarize key findings, but the addition of visual elements like graphs and statistical tables would improve data accessibility. Table 2 may be unnecessary, and its content could be moved to supplementary materials.

Discussion and Conclusions: The discussion is well-structured but could benefit from advice for participants and implications of the findings, such as their influence on dental policies or curriculum reforms. The conclusions are clear and tied to the findings, but they should include suggestions for future studies and a discussion of the study’s limitations.

References and Abstract: The references are up-to-date and relevant, but significant values (e.g., p-values) and MeSH keywords should be added to the abstract for improved searchability.

These revisions will strengthen the manuscript, enhance its readability, and increase its impact on the field of dental education and research.

Reviewer 1 ·

Basic reporting

No Comments

Experimental design

No Comments

Validity of the findings

The paper is written clearly and unambiguously; however, it lacks synthesis of the data and application of this information.

The authors must focus on how the information provided is novel and the knowledge gap needs to be clearly stated.

Additionally, the conclusion needs to be consistent with the data presented.

Reviewer 2 ·

Basic reporting

Language and Clarity:
The manuscript is written in professional, clear, and technically correct English. The language is generally easy to understand, although some sentences in the "Results" section could be made more concise for better clarity.

Background and Context:
The introduction provides adequate background information, including definitions, types, and applications of stem cells. The study highlights the importance of stem cell knowledge among dental faculty and references global and local studies to establish relevance. However, it could further emphasize the direct impact of this knowledge on dental education and research practices.

References:
The references are relevant, up-to-date, and cover a wide range of studies on stem cell therapies. They include local studies to contextualize the research within Saudi Arabia.

Figures and Tables:
The tables effectively summarize key results such as demographic distributions, knowledge scores, and barriers to research. They are well-labeled, clear, and relevant. However, adding visual elements like graphs could enhance the presentation and accessibility of the data.

Raw Data:
The raw data is provided according to the journal’s requirements and aligns with the manuscript’s findings, ensuring transparency.

Experimental design

Research Scope:
The study falls within the scope of the journal by addressing a significant and meaningful topic: the knowledge and attitudes of dental faculty toward stem cell applications. The research question is well-defined and highlights a knowledge gap in this field.

Knowledge Gap:
The manuscript effectively identifies a knowledge and attitude gap among dental faculty and justifies the study by linking it to prior research findings.

Methods:
The methods are described in sufficient detail, allowing replication of the study. The design of the questionnaire, participant recruitment, and data collection procedures are clearly explained. The inclusion of a pilot study strengthens the questionnaire's validity.

Critique1: Cronbach’s Alpha Value Missing:
The study utilized a questionnaire to evaluate knowledge and attitudes, but the Cronbach’s Alpha value was not reported. This value is critical for assessing the internal consistency and reliability of the scales used. Without this information, the reliability of the questionnaire cannot be fully validated. Reporting Cronbach’s Alpha for each subscale would have enhanced the methodological rigor and ensured the credibility of the results derived from the survey.

Critique2: Normality Assumption Testing:
The study used non-parametric tests (Mann-Whitney U and Kruskal-Wallis), which do not require a normality assumption. However, testing for normality (e.g., using the Shapiro-Wilk or Kolmogorov-Smirnov tests) would have provided insights into the data distribution. This would ensure that non-parametric tests were the appropriate choice and provide additional transparency about the dataset.

Ethical Compliance:
The study adheres to ethical standards, with approval from the Institutional Review Board (IRB no. E-22-6999). The authors ensured compliance with the Helsinki Declaration, and informed consent was obtained from participants.

Validity of the findings

Critique1: Cronbach’s Alpha Value Missing:
The study utilized a questionnaire to evaluate knowledge and attitudes, but the Cronbach’s Alpha value was not reported. This value is critical for assessing the internal consistency and reliability of the scales used. Without this information, the reliability of the questionnaire cannot be fully validated. Reporting Cronbach’s Alpha for each subscale would have enhanced the methodological rigor and ensured the credibility of the results derived from the survey.

Critique2: Normality Assumption Testing:
The study used non-parametric tests (Mann-Whitney U and Kruskal-Wallis), which do not require a normality assumption. However, testing for normality (e.g., using the Shapiro-Wilk or Kolmogorov-Smirnov tests) would have provided insights into the data distribution. This would ensure that non-parametric tests were the appropriate choice and provide additional transparency about the dataset.

Replication:
The study is designed to be replicable, as the methodology is detailed, and the questionnaire could be applied to similar populations. The rationale for replication is clear, given the regional variations in knowledge and attitudes.

Conclusions:
The conclusions are directly tied to the findings and are well-stated. The study emphasizes the need for educational efforts and better infrastructure to support stem cell research in Saudi Arabia. However, the authors could elaborate further on how these findings might influence policies or curriculum reforms.

Limitations:
The study acknowledges limitations such as its focus on a single institution and a relatively small sample size. The authors appropriately suggest that future research include broader populations and address additional barriers to stem cell research.

·

Basic reporting

Dear Editor,
Thanks for providing me the opportunity to review the current manuscript. Titled "Knowledge and attitude of dental school faculties towards stem cell therapies and their applications.". The manuscript needs some corrections; please find below the comments to improve the manuscript.
Title:
1. The title is suitable for the study.
Abstract:
1. Kindly write significant values (p values)
2. Kindly add MeSH keywords so the article is easier to search for after publication.
Introduction:
1. The introduction is written shortly; kindly elaborate.
2. Kindly justify the study using literature and elaborate.
3. The problem statement of the study is well written.
4. Kindly add a few studies.
5. Kindly add the hypotheses of the study.

Experimental design

Material and Methods:
1. How was the sample size calculated, and which software was used? Kindly explain.
2. Kindly add the questionnaire in an additional file.
3. Kindly add references.
4. How was the questionnaire calibrated? Are they experts?
Statistical Analysis:
1. Statistical analysis is well conducted and written.

Validity of the findings

Results:
1. The result has been well presented.

Additional comments

Discussion:
1. The discussion is well written.
2. Kindly discuss what advice can be given to the participants.
3. Kindly write briefly about the implementation of the current study.
Limitations:
1. Kindly write all the limitations of the current study.
Conclusions:
1. The conclusion is well written.
2. Kindly write about further study suggestions.
Reference:
1. References are UpToDate.
Tables:
1. Tables no. 1 and 3 are well presented.
2. Kindly add statistical tables, table no. 2 is not required. Add all the data and questioner in an additional file.

---

## Round 0.2 · accepted · Accept

The authors addressed the reviewers' concerns and substantially improved the content of the manuscript. So, based on my own assessment as an editor, no further revisions are required and the manuscript can be accepted in its current form.

Reviewer 2 ·

Basic reporting

The criticisms of the article have been addressed. It is suitable for publication.

Experimental design

The criticisms of the article have been addressed. It is suitable for publication.

Validity of the findings

The criticisms of the article have been addressed. It is suitable for publication.

Additional comments

The criticisms of the article have been addressed. It is suitable for publication.

·

Basic reporting

Nil

Experimental design

Nil

Validity of the findings

Nil

Additional comments

Dear Authors,
The authors have addressed all the comments and suggestions reviewers gave, and the manuscript has dramatically improved. The manuscript can be accepted for publication in its current form. I would like to congratulate the authors and wish them all the very best in their future endeavors.

Best regards and keep well